# Discovery of a Potent PLK1-PBD Small-Molecule Inhibitor as an Anticancer Drug Candidate through Structure-Based Design

**DOI:** 10.3390/molecules24234351

**Published:** 2019-11-28

**Authors:** Yunjiang Zhou, Fang Yan, Xiangyun Huo, Miao-Miao Niu

**Affiliations:** 1Department of Pharmaceutical Analysis, China Pharmaceutical University, Nanjing 210009, China; zyjcqmu@163.com (Y.Z.); lingziyf73@163.com (F.Y.); aowj2015@126.com (X.H.); 2State Key Laboratory of Natural Medicines, School of Basic Medicine and Clinical Pharmacy, China Pharmaceutical University, Nanjing 210009, China

**Keywords:** polo-box domain, pharmacophore modeling, molecular docking, cancer therapy

## Abstract

Polo-box domain of polo-like kinase 1 (PLK1-PBD) has a pivotal role in cell proliferation and could be implicated as a potential anticancer target. Although some small-molecule inhibitors have been developed, their clinical application has been restricted by the poor selectivity. Therefore, there is an urgent need to develop effective PLK1-PBD inhibitors. Herein, we have developed a virtual screening protocol to find PLK1-PBD inhibitors by using combination of structure-based pharmacophore modeling and molecular docking. This protocol was successfully applied to screen PLK1-PBD inhibitors from specs database. MTT assay indicated that five screened hits suppressed the growth of HeLa cells. Particularly, hit-5, as a selective PLK1 inhibitor targeting PLK1-PBD, significantly inhibited the progression of HeLa cells-derived xenograft, with no obvious side effects. This work demonstrates that hit-5 may be a potential anticancer agent.

## 1. Introduction 

Cancer is a highly fatal disease characterized by uncontrolled cell proliferation, leading to the death of 9.6 million people in 2018 [1]. The polo-like kinases (PLKs) are a subfamily of serine-threonine protein kinases that play crucial roles in cellular proliferation [2]. Of the known PLKs, PLK1 is a conserved kinase that plays critical role in cell mitosis [3]. PLK1 participates in a wide range of processes, including mitotic entry, bipolar spindle assembly, centrosome separation, cytokinetic furrow ingression, and dissociation of cohesion from chromosomes [4]. PLK1 has been identified as an attractive target because of its ability to promote tumorigenesis [5]. 

PLK1 is composed of an *N*-terminal catalytic domain and a highly conserved *C*-terminal polo-box domain (PBD) [6]. Over the years, discovery of PLK1 inhibitors has been focused on targeting the catalytic domain. However, protein kinases have high similarities in the ATP-binding sites, and these efforts suffered from a lack of specificity. The PBD of PLK1 has a critical role in proper subcellular localization and mitotic functions of PLK1 by interacting with threonine or serine-phosphorylated peptides [6,7]. Thus, interference with PBD-dependent function induces cell death in vitro and inhibits tumor growth in mouse xenograft models, indicating that PLK1-PBD could be targeted for therapeutic intervention against human cancers [7]. The poor cell permeability and proteolytic instability of the peptides hinder the development of peptide-based inhibitors into novel therapeutic drugs [8]. To overcome these shortcomings of peptide-based inhibitors, several small-molecule compounds including thymoquinone [9], poloxin [10], and poloxipan [11] are identified as Plk1-PBD inhibitors. However, the structural diversity of PLK1-PBD inhibitors and their modest selectivity greatly limit the design based on the structure [12]. Therefore, there is still an urgent need to develop effective therapeutic PLK1-PBD inhibitors. 

Structure-based virtual screening is rapidly becoming a popular alternative to high-throughput screening (HTS) due to it being much cheaper and less time consuming [13]. In previous works, such virtual screening studies were successfully used to identify some PLK1 inhibitors [14,15,16]. In this work, the pharmacophore model was constructed based on the two X-ray crystallographic structures of the PLK1-PBD and further validated using the Gunner–Henry (GH) score method [17]. A pharmacophore-based database search was used to identify potent inhibitors for the PLK1-PBD by a root-mean-square distance (RMSD) value between the query features and their matching ligand annotation points, which is the degree of consistency with the pharmacophore model. Subsequently, these hits were filtered by molecular docking experiments. Finally, five hits were identified as potential leads. 

## 2. Results and Discussion 

### 2.1. Pharmacophore Modeling 

Structure-based drug design (SBDD) and virtual screening have become important tools in the pharmaceutical industry [17,18]. In particular, a general method using same pockets of protein structures to create a receptor-based pharmacophore model plays a vital role in drug discovery [17,18]. The ligand-binding pockets of PLK1-PBD consist of a hydrophobic pocket and a positively charged binding pocket [19,20]. Two crystallographic structures of the PLK1-PBD domain with a high resolution of less than 3 Å were obtained from the Protein Data Bank (PDB) database (Table 1). Compared with other structures such as 4H71, the two ligands (JES107 and Z228588490) of the crystal structures (PDB code: 5NN2 and 5NEI) bound to the same hydrophobic pocket of PLK1-PBD (Appendix A). To identify selective inhibitors targeting the hydrophobic pocket of PLK1-PBD, the two crystal structures were used to generate a structure-based pharmacophore model of the hydrophobic pocket. The generated structure-based model included five features (Figure 1): Two hydrophobic and aromatic features (F1 and F2: Hyd|Aro), two hydrogen bond acceptor features (F3 and F5: Acc), and one hydrogen bond donor feature (F4: Don). As shown in Appendix A, the benzene ring of JES107 or Z228588490 overlaid the hydrophobic and aromatic feature (F1: Hyd|Aro) of pharmacophore, while the other benzene ring of JES107 mapped the hydrophobic and aromatic feature (F2: Hyd|Aro). In addition, the heteroatoms of JES107 and Z228588490 overlaid the hydrogen bond acceptor features (F3 and F5: Acc) and the hydrogen bond donor feature (F4: Don). 

### 2.2. Validation and Database Screening

To confirm the discriminatory ability of the generated pharmacophore model, the model was assessed using the *GH* score as a metric to search a decoy set including 1000 molecules. Some statistical parameters were calculated (Table 2). When a *GH* score is higher than 0.6, the model is very good [17]. It was observed to be 0.77 for the pharmacophore model, suggesting a good ability to distinguish the active from the inactive molecules. 

The flowchart of virtual screening used in this study is displayed in Figure 2. The commercially available specs database consists of 202,919 chemical compounds. Firstly, Lipinski’s rule of drug-likeness derived from the statistics of oral drugs was applied to filter drug-like molecules from the database, owing to the structural characteristics of the PLK1-PBD binding site. Afterward, the validated pharmacophore model was used to identify novel inhibitors from 168,911 drug-like compounds. The RMSD value of 0 indicates the ideal mapping. After virtual screening, 1693 selected hits with an RMSD value less than 0.5 Å were further docked into the PLK1-PBD active site. Then, we used a −7 kcal/mol cutoff in docking score to prune the hit list. The docking scores of five compounds in docking are below −7 kcal/mol. Finally, the five hits (hits 1–5) were selected for biological valuation (Table 3). The five hits show a good pharmacophore mapping on the model (Figure 3). All of the hits were subjected to the pan assay interference compounds (PAINS) online filter (http://cbligand.org/PAINS/) [21]. PAINS analysis showed that five hits passed the filter.

### 2.3. In Vitro Biological Testing 

To evaluate the anticancer activity of these hits, MTT assay was used to test their inhibition to HeLa cells. As shown in Figure 4, five hits inhibited the growth of HeLa cells in a dose-dependent manner. Among these five hits, hit-5 had the best inhibitory effect on the growth of HeLa cells at the concentration of 80 μM (Figure 4), indicating that hit-5 exhibited potent antitumor efficacy. To verify that HeLa cell inhibition effect of hit-5 was dependent on PLK1, the sensitivity of hit-5 to the shControl-HeLa cells and shPLK1 (PLK1-silencing)-HeLa cells was detected by MTT assay. The results showed that the sensitivity of shPLK1-HeLa cells to hit-5 was significantly decreased compared with shControl-HeLa cells (Appendix A), indicating that the HeLa cell inhibition effect of hit-5 is dependent on PLK1.

To further characterize the binding modes of hit-5, we used the microscale thermophoresis (MST) method to measure the binding affinity of hit-5 to the PLKs-PBD. The dissociation constant (*K*_d_) of hit-5 to Plk1-PBD measured by MST was 26 ± 5 μM (Table 4). In addition, the selectivity of hit-5 to PLK1-PBD was higher than that to PLK2-PBD and PLK3-PBD (*K*_d_ > 400). The known PLK1-PBD inhibitors, such as thymoquinone and poloxin, were used as controls. The selectivity ratios of hit-5 to PLK2-PBD and PLK3-PBD (selectivity ratios > 15) are higher than that of thymoquinone and poloxin, indicating that hit-5 is selective for PLK1-PBD. To better understand the molecular interaction between PLK1-PBD and hit-5, it was docked into the active site of PLK1-PBD. The MOE docking experiments revealed that there were two major interactions between hit-5 and the hydrophobic pocket of PLK1-PBD (Figure 5): (i) The *C*-terminal phosphorylated threonine formed multiple hydrogen-bonding interactions with His489 and water molecules that were indispensable for the ligand binding of the PLK1-PBD; (ii) the 9-Ethyl-9*H*-carbazole group of hit-5 was engaged in a strong hydrophobic interaction with the following key residues including Leu478, Phe482, Tyr417, and Tyr485. It has recently been reported that peptide-based inhibitors formed hydrophobic interactions with some key residues in the hydrophobic pocket of PLK1-PBD such as Leu478, Phe482, Tyr417, Tyr485, Tyr481, and Tyr421 [19,20]. Our findings and the previous studies suggest that the interactions of hit-5 are analogous to peptide-based inhibitors of PLK1-PBD. 

### 2.4. In Vivo Tumor Inhibition Ability of Hit-5 

To further investigate the role of hit-5 on tumor progression, we observed the effect of hit-5 on the growth of HeLa cells-derived xenograft in vivo. In the mice experiments, three groups of tumor-bearing mice were injected intraperitoneally with vehicle, hit-5 (10 mg/kg), and hit-5 (50 mg/kg). The results suggested that hit-5 could significantly inhibit the tumor growth (Figure 6a,c,d). In addition, the body weight of the mice treated with hit-5 gradually increased throughout the experiments (Figure 6b), indicating that hit-5 did not have severe side effects. 

## 3. Materials and Methods

### 3.1. Pharmacophore Model Generation and Validation

Two X-ray crystallographic structures of the PLK1-PBD domain with a high resolution of less than 3 Å were obtained from the Protein Data Bank (PDB) database. Firstly, the hydrogen atoms of these protein structures were added using the prepare protein tool within the molecular operating environment (MOE) (Chemical Computing Group Inc, Montreal, Quebec, Canada) and their energy minimizations were performed by the merck molecular force field 94 (MMFF94) force field [22]. On the basis of the chemical properties of the PLK1-PBD active site, hydrogen bond acceptor (Acc), hydrogen bond donor (Don), aromatic center (Aro), and hydrophobic (Hyd) features are further selected for the pharmacophore scheme. Then, these prepared proteins were used for selectively generating the representative features of the PLK1-PBD active site using the pharmacophore query editor protocol of the MOE. The resulting pharmacophore model contains the important pharmacophore features, which represent the essential interaction points with the key residues in the PLK1-PBD active site. 

The Gunner–Henry (GH) scoring method was carried out to verify the quality of the pharmacophore model [17,23]. A decoy set with 30 active molecules obtained from the reported literatures [24,25,26,27] was constructed. Then, the validated model was used as 3D query to filter a decoy set using the pharmacophore search protocol available in MOE. Finally, some statistical parameters statistical parameters were calculated including the total hits (*Ht*), % ratio of actives, % yield of actives, the goodness-of-hit score (*GH*), and enrichment factor (*E*). 

### 3.2. Virtual Screening 

A commercial specs database contains approximately 202,919 chemical compounds. Lipinski’s rule was firstly used to find drug-like molecules from the specs database. Then, a pharmacophore search protocol of the MOE was used to perform virtual screening based on the established pharmacophore model. Hit compounds (hit list) can be ranked according to the root-mean-square distance (RMSD) values between the query features of the model and their matching ligand annotation points [28]. 

### 3.3. Molecular Docking

The crystal structure of PLK1-PBD (PDB ID: 5NN2) was obtained from the PDB database. Hydrogen atoms were added to the protein and energy minimization was performed using the MMFF94 force field. Docking of the ligands into the active site of the PLK1-PBD was performed by the triangle matcher docking protocol of MOE program [28]. The top 30 poses are kept and minimized using MMFF94x within a rigid receptor. The resulting poses are then scored using the dG docking scoring function [28,29]. 

### 3.4. Microscale Thermophoresis (MST) Analysis 

According to a previously reported method [27], the PBD of PLKs was l labeled using the Lys labeling kit for detection in the MST experiments with the red fluorescent dye NT-647. Measurements were carried out in the solution containing 20 mM Tris (pH 8.0), 200 mM NaCl, 0.1% (v/v) TWEEN-20, 1 mM EDTA, and 0.1 mg/mL BSA, by using 20% LED power. The concentration of PBD was adjusted by the fluorescence counts. The tested compounds dilution series and PBD solution were mixed 1:1 to obtain the final measurement samples. DMSO was used as negative control. Nanotemper Analysis software V1.5.41 was applied to analyze data. 

### 3.5. MTT Cell Proliferation Assay, Transfection of PLK1 shRNA and Western Blot Assay

Human HeLa cervical cancer cells were obtained from Shanghai Cell Bank of the Chinese Academic of Sciences (Shanghai, China). Cells were seeded in 96-well culture plate and allowed to grow overnight. Then, cells were exposed to different concentrations of hit-5 and incubated at 37 °C for 72 h. After that, a MTT stock solution (0.5 mg/mL) was added into each well and the plate was incubated for 4 h. The MTT-treated cells were fixed with 150 µL of DMSO. The absorbance of each individual well was measured at 490 nm on a microplate spectrophotometer. 

For transfection of shRNA, lentiviral particles encoding non-target shRNA and PLK1 shRNA were diluted in OptiMEM containing 6 μg/mL of polybrene, and then were added to HeLa cells. After 3 days, 5 μg/mL of puromycin were used to select transfected cells. Cells transfected with the shRNA lentiviral particles were seeded into six-well plates and Western blot analysis was used to detect the protein levels of PLK1. 

For Western blot assay, HeLa cells were washed and lysed with RIPA. Protein samples were detected by Western blot using a standard protocol. Anti-actin antibody was used as loading control antibody for total protein samples. The relative protein level of PLK1 was analyzed by using Image J software (National Institutes of Health, Bethesda, MD, USA). 

### 3.6. In Vivo Anticancer Activity

The experiments involving animals were approved by the Animal Ethics Committee of China Pharmaceutical University (ethic approval number: 2019-10-003). Fifteen female nude mice (BALB/c, 6 weeks old) were purchased from the Experimental Animal Center of Yangzhou University. HeLa cells (200 μL, 10^7^ cells) were injected subcutaneously into the flank. Once tumors grew to 80–100 mm^3^, mice were randomly assigned into three groups and injected intraperitoneally with vehicle, hit-5 (10 mg/kg), and hit-5 (50 mg/kg). Tumor volume and body weight were measured every 3 days. Tumor volume was calculated using the formula (c × c × d)/2 (c, the smallest diameter; d, the largest diameter). 

## 4. Conclusions

In summary, we have successfully constructed a virtual screening protocol including pharmacophore modeling and molecular docking. The potent hit-5 obtained from specs database can significantly inhibit the growth of human cervical cancer HeLa cells. The hit-5 may facilitate to identify and optimize new leads for PLK1-PBD inhibition. In addition, the protocol can also be used for virtual screening of other chemical databases to identify potent PLK1-PBD inhibitors with unknown scaffolds. 

## Figures and Tables

**Figure 1 molecules-24-04351-f001:**
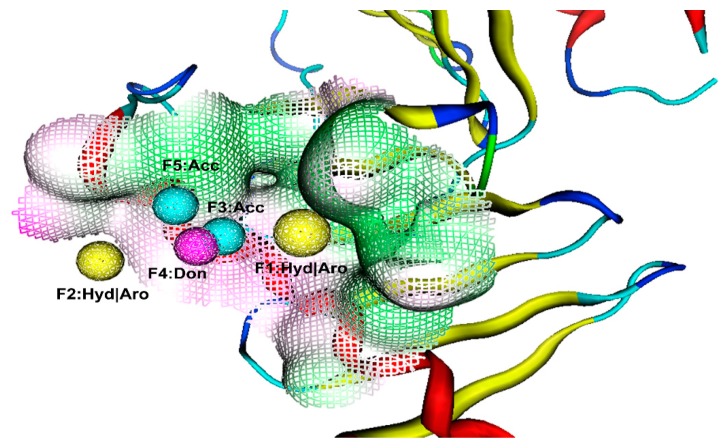
The generated pharmacophore model in the binding site of PLK1-PBD. Pharmacophore features are color-coded: Yellow, two hydrophobic and aromatic features (F1 and F2: Hyd|Aro); cyan, two hydrogen bond acceptor features (F3 and F5: Acc); purple, one hydrogen bond donor feature (F4: Don). The protein backbone is shown in tube form; a reticulate pocket represents the shape of the binding site in PLK1-PBD.

**Figure 2 molecules-24-04351-f002:**
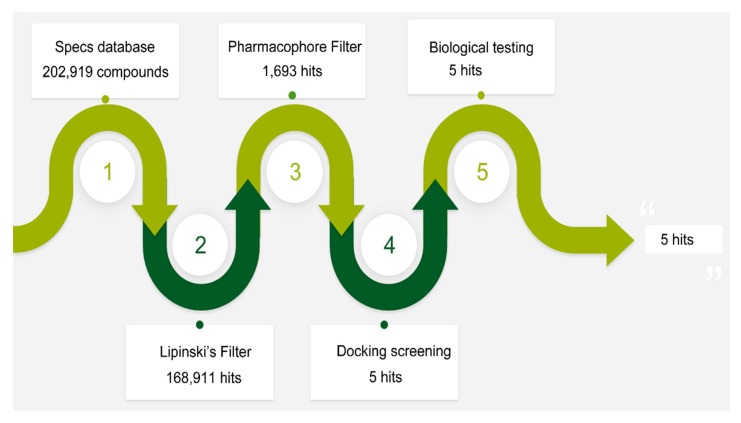
A workflow overview of pharmacophore modeling, selection of compounds and biological testing.

**Figure 3 molecules-24-04351-f003:**
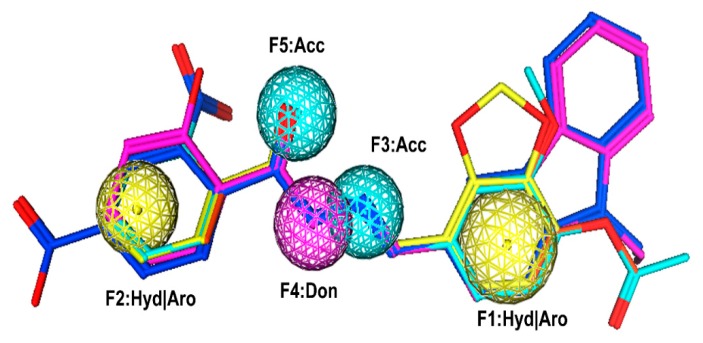
Pharmacophore mapping of five hits on the model. Pharmacophore features are color-coded: Yellow, two hydrophobic and aromatic features (F1 and F2: Hyd|Aro); cyan, two hydrogen bond acceptor features (F3 and F5: Acc); purple, one hydrogen bond donor feature (F4: Don). The hits are shown in stick form.

**Figure 4 molecules-24-04351-f004:**
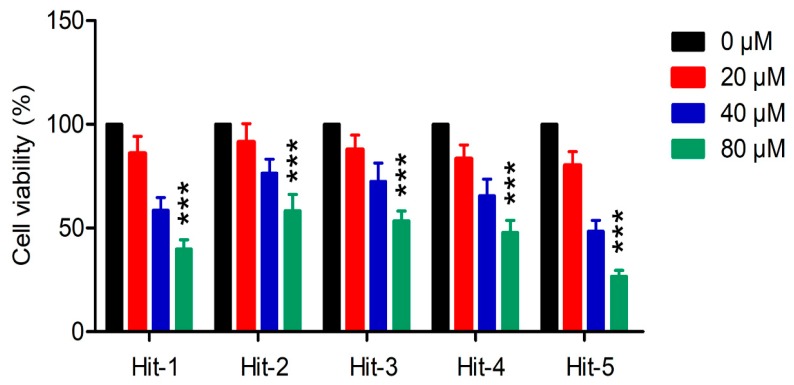
Growth inhibition effects of selected hits 1–5 on HeLa cells. The results are representative of three independent experiments and are expressed as mean ± SD. *** *P* < 0.001.

**Figure 5 molecules-24-04351-f005:**
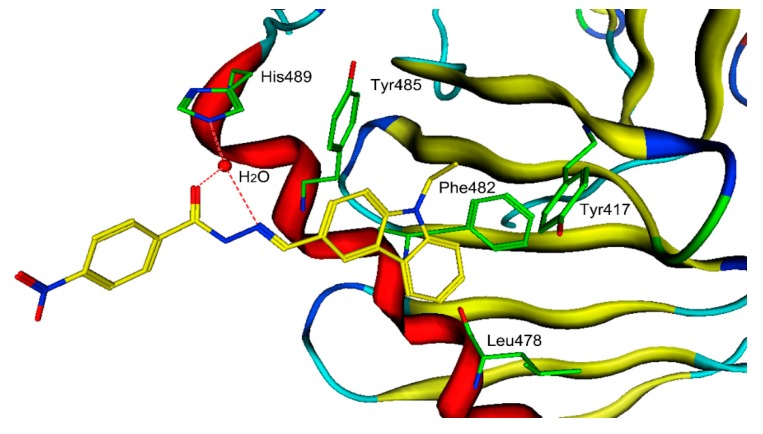
The three-dimensional (3D) ligand–protein interaction diagram for the binding site of PLK1-PBD (PDB ID: 5NN2) with hit-5. The active site residues are shown in green stick form. Hit-5 is color-coded by yellow. The hydrogen-bond network with protein residues is represented by red dotted lines. The protein backbone is shown in tube form.

**Figure 6 molecules-24-04351-f006:**
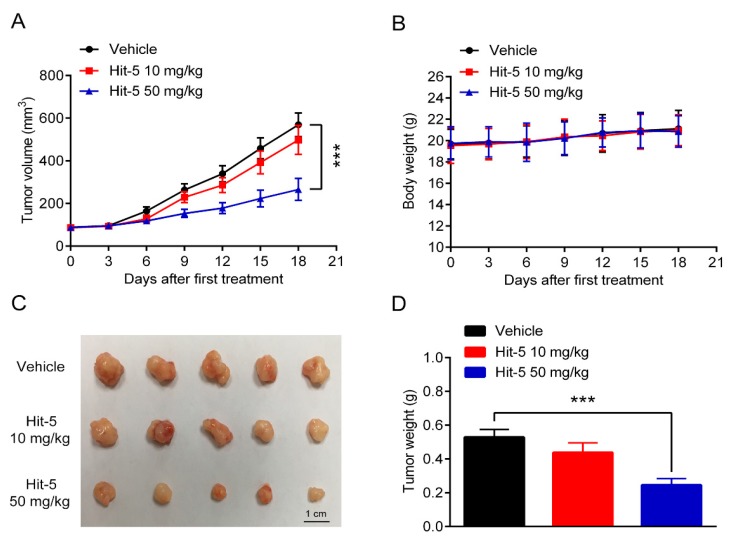
In Vivo antitumor activities of hit-5. (**A**) The tumor volume changes. (**B**) The body weight changes. (**C**) The image of tumors. (**D**) The tumor weights of each group. Data are represented as mean ± SD, n = 5. *** *P* < 0.001.

**Table 1 molecules-24-04351-t001:** Basic information of receptor-ligand complexes of the PLK1-PBD domain from the Protein Data Bank (PDB) database.

PDB_ID	Resolution (Å)	Ligand_Sequence
5NN2	1.81	Z24
5NEI	2.68	8VB

**Table 2 molecules-24-04351-t002:** Pharmacophore model validation by goodness-of-hit score (*GH*) score method.

Serial No.	Parameter	Pharmacophore Model
1	Total molecules in database (*D*)	1000
2	Total number of actives in database (*A*)	30
3	Total hits (*Ht*)	34
4	Active hits (*Ha*)	25
5	% Yield of actives [(*Ha*/*Ht*) × 100]	74
6	% Ratio of actives [(*Ha*/*A*) × 100]	83
7	Enrichment factor (*E*) [(*Ha* × *D*)/(*Ht* × *A*)]	25
8	False negatives [*A* − *Ha*]	5
9	False positives [*Ht* − *Ha*]	9
10	Goodness of hit score (*GH*) ^a^	0.77

^a^ (*Ha*(3*A* + *Ht*)/(4*HtA*))(1 − (*Ht* − *Ha*)/(*D* − *A*)); *GH* score of more than 0.6 indicates a good model.

**Table 3 molecules-24-04351-t003:** Results of root-mean-square distance (RMSD) values and docking scores of five selected hits.

Hits	ID Number	Structure	RMSD [Å] ^(a)^	Docking Score [kcal/mol] ^(b)^	Pharmacophore/Docking Ranking ^(c)^
1	AN-329/40093808	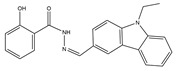	0.3152	–7.6527	6/2
2	AG-205/07764017	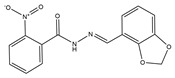	0.3668	–7.2534	13/4
3	AN-329/10002015	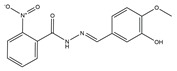	0.3609	–7.2088	16/5
4	AN-329/10080005	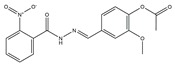	0.3401	–7.4651	11/3
5	AN-329/10077003	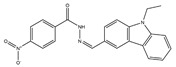	0.3128	–8.0127	5/1

^(a)^ The root-mean-square distance between the query features and their matching ligand annotation points; ^(b)^ Binding free energy between PLK1-PBD and a ligand; ^(c)^ Pharmacophore ranking and molecular docking ranking of these hits out of total screened for each method.

**Table 4 molecules-24-04351-t004:** The binding affinity of hit-5 to PLKs-polo-box domain (PBD).

Compounds	PLK1-PBD [μM] ^(a)^	PLK2-PBD [μM] ^(a)^	PLK3-PBD [μM] ^(a)^	PLK2-PBD/PLK1-PBD	PLK3-PBD/PLK1-PBD
Hit-5	26 ± 5	>400	>400	>15	>15
Poloxin	34 ± 7	152 ± 11	284 ± 17	4.5	8.4
Thymoquinone	39 ± 6	46 ± 5	55 ± 10	1.2	1.4

^(a)^ The results are representative of three independent experiments and are expressed as mean ± SD.

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
