# Peer review of "Discovery of a Potent PLK1-PBD Small-Molecule Inhibitor as an Anticancer Drug Candidate through Structure-Based Design"

_molecules, 2019, doi:10.3390/molecules24234351_

Round 1

Reviewer 1 Report

In this manuscript, authors presented the discovery of small molecule inhibitors of PLK1-PBD. Authors have employed virtual screening using Lipinski filter, pharmacophore screening and molecular docking to select five hits that were tested experimentally. Overall, I feel there are several shortcomings in the manuscript and should only be published in Molecules after addressing following comments:

Abstract: “structure-based pharmacophore modeling, virtual screening and molecular docking..” should be structure-based pharmacophore modeling and molecular docking. Abstract: “hit-5 may be a potential anticancer drug” It is a big claim without evidences. Should be changed to anticancer compound/agent.  Page 2, Line 51, “Gunner-Henry(GH)..”. Citation should be provided. Page 2, Line 52, “root-mean-square distance”.. RMSD with what? Page 2, Line 59, “ a novel method…” development of consensus pharmacophore model is not novel. This sentence should be rephrased. Page 2, Figure 1, another panel showing the overlay of 5NN2 and 5NEI crystal ligands with pharmacophore features would be nice. Moreover, it should be described in text why these features were selected. Whether the selected pharmacophore features are consensus features from two query crystal ligands? While screening for compounds with these features, whether the presence all features is required for a hit or not? Page 3, Line 78, “test set containing 992 inactive and 8 active.” How is test set was created. I believe 992 inactive may be just property matched decoys and not experimentally tested. So, the use of term decoys may be more suitable. Why only 8 active hits were selected? I believe there are more PLK1-PDB inhibitors in literature. Page 3, Line 81, “When a GH score is higher than 0.6” If this is observation from previous study, citation should be provided. Page 3, Line 96, “Finally, five hits.” How the se 5 hits were selected? Whether these are top scoring compounds in docking? Even if these compounds were selected in cerebro, explanation should be provided. This is necessary as all hits belong to one chemical class and it give reader an impression that virtual screening study was designed to select just these compounds. Page 4, Table 3, Pharmacophore ranking and molecular docking ranking of these hits out of total screened for each method should be provided. Page 4, 5, 6. Assay results. Biggest shortcoming of this manuscript is that there is only one data (MST of hit-5) that shows direct target engagement. Apartment from this there is no evidence that these compounds are PLK1-PBD inhibitors. Moreover, as Kd value of hit-5 is very weak, other results, such as anticancer activity and in vivo tumor suppression may be either due to promiscuous nature of compounds or engaging other targets. This is a big concern as all hits contain acyl hydrazone which is classified as PAINS (Baell et al. J. Med. Chem. 2010, 53, 2719–2740) Another big shortcoming of this manuscript is the selectivity of reported compounds. Authors goal was to design selective PLK1-PBD domain inhibitors as PLK1 catalytic domain inhibitors are non-selective due to active site similarity. However, selectivity of reported compounds was not evaluated against other kinases. Table 4 shows selectivity against other PLK isoforms, however, N.D. doesn’t mean that compounds are selective. Page 8, Section 3.6 In vivo anticancer activity, How many mice were used in this experiment?

Author Response

Response to Reviewer 1 Comments

Point 1: English language and style

( ) Extensive editing of English language and style required

( ) Moderate English changes required

(x) English language and style are fine/minor spell check required

( ) I don't feel qualified to judge about the English language and style   

Response 1: Thank you for the suggestion. We have carefully corrected some grammatical errors in the revised manuscript.   

Point 2: Overall, I feel there are several shortcomings in the manuscript and should only be published in Molecules after addressing following comments:

Abstract: “structure-based pharmacophore modeling, virtual screening and molecular docking.” should be structure-based pharmacophore modeling and molecular docking.

Response 2: Thank you for the suggestion. According to your advice, the mistake has been corrected. The related description has been added in the revised manuscript (please see “Abstract”).   

Point 3: Abstract: “hit-5 may be a potential anticancer drug” It is a big claim without evidences. Should be changed to anticancer compound/agent. 

Response 3: Thank you for the suggestion. The related spelling error has been corrected in the revised manuscript (please see “Abstract”). 

Point 4: Page 2, Line 51, “Gunner-Henry(GH)..”. Citation should be provided.

Response 4: Thank you for the suggestion. The related citation has been added in the revised manuscript (please see “1. Introduction and Reference 17”).   

Point 5: Page 2, Line 52, “root-mean-square distance”. RMSD with what?

Response 5: Thank you for the suggestion. The related explanation of RMSD has been added in the revised manuscript (please see “1. Introduction”).

Point 6: Page 2, Line 59, “a novel method…” development of consensus pharmacophore model is not novel. This sentence should be rephrased.

Response 6: Thank you for the suggestion. This sentence has been rephrased. The related description has been added in the revised manuscript (please see “2.1. Pharmacophore modeling”).      

Point 7: Page 2, Figure 1, another panel showing the overlay of 5NN2 and 5NEI crystal ligands with pharmacophore features would be nice. Moreover, it should be described in text why these features were selected. Whether the selected pharmacophore features are consensus features from two query crystal ligands? While screening for compounds with these features, whether the presence all features is required for a hit or not?     

Response 7: Thank you for the suggestion. As shown in Figure S2, the benzene rings of JES107 and Z228588490 overlaid one hydrophobic and aromatic feature (F1: Hyd|Aro) of pharmacophore, while the other benzene ring of JES107 mapped one hydrophobic and aromatic feature (F2: Hyd|Aro). In addition, the oxygen or sulfur atoms of JES107 and Z228588490 overlaid two hydrogen bond acceptor features (F3 and F5: Acc) and one hydrogen bond donor feature (F4: Don). The results indicated that the pharmacophore model may be utilized to screen the compounds that have a chemical structure with functional groups closely related to those of JES107 or Z228588490. While screening for compounds with these features, the presence all features is required for a hit. As shown in Figure 3, five hits can overlay all features of the pharmacophore model. The related description has been added in the revised manuscript (please see “2.2. Validation and database screening and Figure S2”).      

Point 8: Page 3, Line 78, “test set containing 992 inactive and 8 active.” How is test set was created. I believe 992 inactive may be just property matched decoys and not experimentally tested. So, the use of term decoys may be more suitable.

Response 2: Thank you for the suggestion. The decoy set has been added in the revised manuscript (please see “2.2. Validation and database screening”).     

Point 8: Why only 8 active hits were selected? I believe there are more PLK1-PDB inhibitors in literature.

Response 2: Thank you for the suggestion. We select 30 active hits collected from the reported literatures to assess the GH score again. The related results have been added in the revised manuscript (please see “Table 2 and References [24–27]”).     

Point 9: Page 3, Line 81, “When a GH score is higher than 0.6” If this is observation from previous study, citation should be provided.

Response 9: Thank you for the suggestion. The related citation has been added in the revised manuscript (please see “2.2. Validation and database screening and Reference 17”).

Point 10: Page 3, Line 96, “Finally, five hits.” How the se 5 hits were selected? Whether these are top scoring compounds in docking? Even if these compounds were selected in cerebro, explanation should be provided. This is necessary as all hits belong to one chemical class and it give reader an impression that virtual screening study was designed to select just these compounds.

Response 10: Thank you for the suggestion. Considering a cutoff to classify compounds as active and inactives, we used a -7 kcal mol-1 cutoff in docking score to prune the hit list (lower values indicate better binding affinity). The docking scores of only 5 top scoring compounds in docking are below -7 kcal mol-1. Finally, the 5 hits (hits 1-5) were selected for biological testing (Table 3). Figure 3 depicts a good pharmacophore mapping of five hits on the model. All of the hits were subjected to the pan assay interference compounds (PAINS) online filter (http://cbligand.org/PAINS/). PAINS analysis showed that five hits passed the filter. According to your advice, the related explanation have been added in the revised manuscript (please see “2.2. Validation and database screening”).    

Point 11: Page 4, Table 3, Pharmacophore ranking and molecular docking ranking of these hits out of total screened for each method should be provided.

Response 11: Thank you for the suggestion. Pharmacophore ranking and molecular docking ranking have been added in the revised manuscript (please see “Table 3”).  

Point 12: Page 4, 5, 6. Assay results. Biggest shortcoming of this manuscript is that there is only one data (MST of hit-5) that shows direct target engagement. Apartment from this there is no evidence that these compounds are PLK1-PBD inhibitors. Moreover, as Kd value of hit-5 is very weak, other results, such as anticancer activity and in vivo tumor suppression may be either due to promiscuous nature of compounds or engaging other targets. This is a big concern as all hits contain acyl hydrazone which is classified as PAINS (Baell et al. J. Med. Chem. 2010, 53, 2719–2740).

Response 12: Thank you for the suggestion. In previous studies, we have constructed shControl and shPLK1 (PLK1-silencing)-HeLa cells (A Redox-Triggered Bispecific Supramolecular Nanomedicine Based on Peptide Self-Assembly for High-Efficacy and Low-Toxic Cancer Therapy, https://doi.org/10.1002/adfm.201904969). To verify that HeLa cell inhibition effect of hit-5 was dependent on PLK1, the sensitivity of hit-5 to the shControl-HeLa cells and shPLK1-HeLa cells was detected by MTT assay. The results showed that the sensitivity of shPLK1-HeLa cells to hit-5 was significantly decreased compared with shControl-HeLa cells (Figure S3, Supporting Information), indicating that HeLa cell inhibition effect of hit-5 was dependent on PLK1. In addition, all of the hits (hits 1-5) were subjected to the pan assay interference compounds (PAINS) online filter (http://cbligand.org/PAINS/). PAINS analysis showed that five hits passed the filter. Related results and description have been added to the revised manuscript and supporting information (please see “Figure S3, Table 4 and 2.3. In vitro biological testing”).

Point 13: Another big shortcoming of this manuscript is the selectivity of reported compounds. Authors goal was to design selective PLK1-PBD domain inhibitors as PLK1 catalytic domain inhibitors are non-selective due to active site similarity. However, selectivity of reported compounds was not evaluated against other kinases. Table 4 shows selectivity against other PLK isoforms, however, N.D. doesn’t mean that compounds are selective.  

Response 13: Thank you for the suggestion. To evaluate the selectivity of hit-5 against PLK2-PBD and PLK3-PBD, the microscale thermophoresis (MST) method was performed. As shown in Table 4, the selectivity of hit-5 to PLK1-PBD was higher than that to PLK2-PBD and PLK3-PBD (Kd > 400). The known PLK1 inhibitors such as thymoquinone and poloxin were used as controls. The selectivity ratios of hit-5 to PLK2-PBD and PLK3-PBD (selectivity ratios > 15) are higher than that of thymoquinone and poloxin, indicating that hit-5 is selective for PLK1-PBD. The results and related description have been added in the revised manuscript (please see “Table 4 and 2.3. In vitro biological testing”).           

Point 14: Page 8, Section 3.6 In vivo anticancer activity, How many mice were used in this experiment?

Response 14: Thank you for the suggestion. Fifteen female nude mice (BALB/c, 6 weeks old) were purchased from the Experimental Animal Center of Yangzhou University. The related description has been added in the revised manuscript (please see “3.6. In vivo anticancer activity”).  

Reviewer 2 Report

Zhou et. al., identified PLK1-PKD inhibitors via virtual screening. Compounds identified were subsequently shown to be active in an MTT assay. While the results are promising, the article may not be suitable for publication in current form for a good impact factor journal like Molecules. Hence, I recommend the article for major revision.

Some questions and comments - 

One of the major concerns is the PDBs used in the virtual screening study. It is not clear why authors chose 5NN2 and 5NEI. I couldn't find any paper published with the associated PDBs.
Why didn't the authors choose other available PDBs like 4H71? Are there any regions near the binding site that is unresolved in the PDBs used in modelling studies? If so, how did you handle unresolved regions? Did you also try docking compounds to other PBDs like 4H71? Are there any major differences noticed? One way to generate confidence in the study is to overlay 5NN2, 5NEI vs. PDB like 4H71 and make sure there are no major differences with respect to the site of interest.

What is the rational behind using 2 PDB to generate pharmacophores? What are the similarities and differences between the two crystal structures? Recent advances in computer-aided drug design CTMC, 2014, 14,1875 discusses about cases where 'ensemble docking' is applicable.

Introduction should include review of known PLK1 inhibitors and previous virtual screening studies.

How do we know hit-5 doesn't show binding to PLK2-PBD and PLK3-PBD?

Are the interactions of hit-5 analogous to known inhibitors of PLK1? Docked pose needs justification with reference to previous experimental or crystal studies.

Author Response

Response to Reviewer 2 Comments

Point 1: English language and style

( ) Extensive editing of English language and style required

( ) Moderate English changes required

(x) English language and style are fine/minor spell check required

( ) I don't feel qualified to judge about the English language and style

Response 1: Thank you for the suggestion. We have carefully corrected some grammatical errors in the revised manuscript. 

Point 2: Some questions and comments:

One of the major concerns is the PDBs used in the virtual screening study. It is not clear why authors chose 5NN2 and 5NEI. I couldn't find any paper published with the associated PDBs.

Why didn't the authors choose other available PDBs like 4H71? Are there any regions near the binding site that is unresolved in the PDBs used in modelling studies? If so, how did you handle unresolved regions? Did you also try docking compounds to other PBDs like 4H71? Are there any major differences noticed? One way to generate confidence in the study is to overlay 5NN2, 5NEI vs. PDB like 4H71 and make sure there are no major differences with respect to the site of interest.   

Response 2: Thank you for the suggestion. According to your suggestion, the protein superpose protocol of the MOE program was used to overlay the three crystal structures of PLK1-PBD (PDB code: 5NN2, 5NEI and 4H71). As shown in Figure S1, ligand-binding pockets of PLK1-PBD consist of a hydrophobic pocket and a positively charged binding pocket (Angew. Chem., Int. Ed. 2012, 51, 10078-10081; Angew. Chem., Int. Ed. 2011, 50, 4003-4006). Figure S1 describes binding pockets of PLK1-PBD as revealed by small-molecule ligands. We can see from Figure S1 that two ligands JES107 and Z228588490  bound to the left hydrophobic pocket derived from two crystal structures of PLK1-PBD (PDB code: 5NN2 and 5NEI), while the ligand thymoquinone (namely, poloxime) only bound to the right charged binding pocket from the crystal structure (PDB code: 4H71). Although some PLK1-PBD inhibitors including thymoquinone and poloxin targeting the positively charged binding pocket have been reported, many of them have been limited by their poor selectivity. Targeting the hydrophobic pocket of PLK1-PBD helps avoid the selectivity issue of these inhibitors. The MOE program can create pharmacophore features by overlapping two ligands binding to the same pocket derived from crystal structures of a single specific receptor (Manual of molecular operating environment (MOE), Version 2007.09). Therefore, to identify new selective PLK1-PBD inhibitors targeting the hydrophobic pocket, we choose 2 PDB (PDB code: 5NN2 and 5NEI) to generate a pharmacophore model of the hydrophobic pocket of PLK1-PBD by using the MOE program (Figure S2). As shown in table 4, the selectivity ratios of the screened hit-5 to PLK2-PBD and PLK3-PBD (selectivity ratios > 15) are much higher compared with known PLK1-PBD inhibitors thymoquinone and poloxin, indicating that hit-5 is selective for PLK1-PBD. The results and related description have been added in the revised manuscript and supporting information (please see “2.1. Pharmacophore modeling and Figure S1,S2”).                 

Figure S1. The overlay of three crystal structures (PDB code: 5NN2, 5NEI and 4H71). Binding pockets of PLK1-PBD as revealed by small-molecule ligands are shown in gray. Crystal structures (PDB code: 5NN2 and 5NEI) of PLK1-PBD in complex with two ligands (Z228588490 and JES107) bound to the hydrophobic pocket (on the left side). Crystal structure (PDB code: 4H71) of PLK1-PBD in complex with a ligand (thymoquinone) bound to the positively charged binding pocket (on the right side). Z228588490, JES107 and thymoquinone are shown in sticks of orange, purple and blue, respectively.

Point 3: What is the rational behind using 2 PDB to generate pharmacophores? What are the similarities and differences between the two crystal structures? Recent advances in computer-aided drug design CTMC, 2014, 14,1875 discusses about cases where 'ensemble docking' is applicable.           

Response 3: Thank you for the suggestion. Structure-based pharmacophores were generated using Molecular Operating Environment (MOE) (Chemical Computing Group Inc, Montreal, Quebec, Canada). The MOE program can create pharmacophore features by overlapping two ligands binding to the same pocket derived from crystal structures of a single specific receptor (Manual of molecular operating environment (MOE), Version 2007.09). Therefore, to construct a reasonable pharmacophore model directly from the same pocket of complex structures, the selection of crystal structures for generating pharmacophores should be according to the following criteria: (i) all the crystal structures of a single specific target have the same amino acid sequence without deletion and error of residues and crystal structures is at best obtained from the same laboratory; (ii) ligands bind to the same pocket derived from crystal structures (Manual of molecular operating environment (MOE), Version 2007.09).       

Here are several key reasons for using the 2 PDB (PDB code: 5NN2 and 5NEI) to generate pharmacophores. Firstly, compared with other structures such as 4H71, two crystal structures of PLK1-PBD (PDB code: 5NN2 and 5NEI) is obtained from the same laboratory, thus the number and length of their amino acid sequences are almost identical, and their binding-ligand pocket have the same size (Figure S2). In addition, two crystal structures have not deletion and error of residues. Secondly, as shown in Figure S2, the two ligands (JES107 and Z228588490) from two crystal structures (PDB code: 5NN2 and 5NEI) bind to the same hydrophobic pocket of PLK1-PBD. To develop a pharmacophore model of the hydrophobic pocket of PLK1-PBD, we chose the two crystal structures (PDB code: 5NN2 and 5NEI) of ligands bound to the same hydrophobic pocket. Thirdly, the conservative structure of the positively charged binding pocket lead to the selectivity issue of PLK1-PBD inhibitors such as poloxin and poloxipan, thus development of inhibitors targeting the hydrophobic pocket becomes a new choice. Based on the above considerations, we chose the two crystal structures to generate a pharmacophore model of the hydrophobic pocket of PLK1-PBD by using the MOE program. The results and related description have been added in the revised manuscript (please see “Figure S2 and 2.1. Pharmacophore modeling”).            

Point 4: Introduction should include review of known PLK1 inhibitors and previous virtual screening studies. 

Response 4: Thank you for the thoughtful suggestion. The review of known PLK1 inhibitors and previous virtual screening studies have been added in the revised manuscript (please see “Introduction”).

Point 5: How do we know hit-5 doesn't show binding to PLK2-PBD and PLK3-PBD?

Response 5: Thank you for the thoughtful suggestion. To further confirm the selectivity of hit-5, we used the microscale thermophoresis (MST) method to measure the binding affinity of hit-5 to PLK2-PBD and PLK3-PBD. The results indicated that the selectivity of hit-5 to PLK1-PBD were higher than that to PLK2-PBD and PLK3-PBD (Kd > 400). The known PLK1-PBD inhibitors including thymoquinone and poloxin were used as controls. As shown in Table 4, the selectivity ratios of hit-5 to PLK2-PBD and PLK3-PBD (selectivity ratios > 15) are much higher than that of thymoquinone and poloxin. The results and related description have been added in the revised manuscript (please see “Table 4 and 2.3. In vitro biological testing”).                

Point 6: Are the interactions of hit-5 analogous to known inhibitors of PLK1? Docked pose needs justification with reference to previous experimental or crystal studies.  

Response 6: Thank you for the thoughtful suggestion. We have compared the interactions of hit-5 with known inhibitors of PLK1. It has recently reported that some peptide-based inhibitors formed hydrophobic interactions with some key residues in the hydrophobic pocket of PLK1-PBD such as Leu478, Phe482, Tyr417, Tyr485, Tyr481 and Tyr421 (Angew. Chem., Int. Ed. 2011, 50, 4003-4006; Angew. Chem., Int. Ed. 2012, 51, 10078-10081). As shown in Figure 5, the 9-Ethyl-9H-carbazole group of hit-5 was engaged in a strong hydrophobic interaction with the following key residues including Leu478, Phe482, Tyr417 and Tyr485. Our results and the previous studies suggest that the interactions of hit-5 are analogous to peptide-based inhibitors of PLK1-PBD. The related reference and description have been added in the revised manuscript (please see “2.3. In vitro biological testing, Figure 5 and References 19 and 20”).                

Round 2

Reviewer 1 Report

All the comments raised by this reviewer were adequately addressed. I have no further comments.

Author Response

Response to Reviewer 1 Comments

Point 1: English language and style

( ) Extensive editing of English language and style required 
( ) Moderate English changes required 
(x) English language and style are fine/minor spell check required 
( ) I don't feel qualified to judge about the English language and style  

Response 1: Thank you for the suggestion. We have carefully corrected some grammatical errors in the revised manuscript.   

Point 2:All the comments raised by this reviewer were adequately addressed. I have no further comments.

Response 2: Thank you very much for your advice! This is particularly important for improving the quality of our paper.

Reviewer 2 Report

Zhou et. al., addressed most of the questions. The article is hence recommended for publication after addressing the following issues.

There is still scope to improve introduction by including earlier virtual screening studies on PLK1. I still do not understand the rational behind using 2 similar PDBs in pharmacophore generation. 

Author Response

Response to Reviewer 2 Comments

Point 1: English language and style

( ) Extensive editing of English language and style required 
( ) Moderate English changes required 
(x) English language and style are fine/minor spell check required 
( ) I don't feel qualified to judge about the English language and style

Response 1: Thank you for the suggestion. We have carefully corrected some grammatical errors in the revised manuscript. 

Point 2: Zhou et. al., addressed most of the questions. The article is hence recommended for publication after addressing the following issues.

There is still scope to improve introduction by including earlier virtual screening studies on PLK1. I still do not understand the rational behind using 2 similar PDBs in pharmacophore generation.    

Response 2: Thank you for the suggestion. According to your suggestion, the earlier virtual screening studies on PLK1 were added to the introduction of the revised manuscript (please see “1. Introduction and References [14-16]”). In previous works, both complexes have successfully served as starting points for model generation. Some researchers clarifies validity and rationality of using two similar complexes in pharmacophore generation. For example, schuster et al recently reported that entries from the Protein Data Bank (PDB) 1equ and 1i5r, two complexes of 17β-HSD1 with the cofactor NADP+ and an inhibitor, served as starting points for model generation (J. Med. Chem. 2008, 51, 4188–4199). Virtual screening with the pharmacophore model was used to identify nonsteroidal 17β-HSD1 inhibitor scaffolds. Analysis of 14 selected compounds yielded four that inhibited the activity of human 17β-HSD1 (IC50 below 50 µM);  Rollinger et al reported that as a starting point for inhibitor modeling, two cocrystallization complexes of HRV with the ligands (PDB ID: 1QJU and 1C8M) were chosen (J. Med. Chem. 2008, 51, 842–851). The model was designed to combine features characteristic for ligand binding in the hydrophobic pocket. Through computational structure-based screening of an in-house 3D database containing 9676 individual plant metabolites from ancient herbal medicines, sesquiterpene coumarins from the gum resin asafetida were selected as promising natural products. The above research confirms the applicability and potency of the pharmacophore model for the rational search for drug leads, enabling a fast and efficient identification of potential new inhibitors of the target protein.                     
